# Ring-Finger Protein 126 (RNF126) Promotes Anoikis Resistance and Peritoneal Colonization in Ovarian Cancer

**DOI:** 10.3390/ijms262412183

**Published:** 2025-12-18

**Authors:** Anh Duc Vu, Shiori Mori, Kanako Akamatsu, Jun Nakayama, Takeharu Sakamoto

**Affiliations:** 1Department of Cancer Biology, Institute of Biomedical Science, Kansai Medical University, Osaka 573-1010, Japan; 2Department of Oncogenesis and Growth Regulation, Research Institute, Osaka International Cancer Institute, Osaka 541-8567, Japan

**Keywords:** anoikis, NF-κB, peritoneal dissemination, ovarian cancer, RNF126

## Abstract

Ovarian cancer (OC) represents the most lethal gynecologic malignancy because the majority of patients with OC are diagnosed at advanced stages with peritoneal colonization of OC cells owing to subtle and nonspecific nature of symptoms. Thus, peritoneal colonization-directed therapeutic approaches are urgently needed for patients with advanced OC. Here, we investigated whether Ring-finger protein 126 (RNF126), an E3 ubiquitin ligase that is aberrantly upregulated in epithelial OC tissues, contributes to the peritoneal colonization of OC. RNF126-depleted OC cells showed comparable proliferation under normal culture conditions but displayed decreased growth under floating (anchorage-independent) conditions in vitro. Further analyses showed that RNF126 promoted anoikis resistance in vitro and increased peritoneal colonization in immunodeficient mice in a RING domain-dependent manner. Mechanistically, RNF126 activated the transcriptional factor NF-κB in OC cells under floating conditions in a RING domain-dependent manner, and this NF-κB activation was essential for anchorage-independent growth and peritoneal colonization of OC cells. Thus, RNF126 is a possible target for the prevention and/or therapy of peritoneally colonized OC.

## 1. Introduction

Ovarian cancer (OC) represents the most lethal gynecologic malignancy, claiming lives of over 200,000 women annually worldwide. It exhibits poor prognosis, with five-year survival rates remaining below 40% [1,2]. The majority of patients with OC are diagnosed at advanced stages of peritoneal metastasis owing to subtle and nonspecific nature of symptoms. More than 70% of patients with OC present with International Federation of Gynecology and Obstetrics (FIGO) stage III/IV disease at initial diagnosis, which significantly compromises the efficacy of surgical and radiotherapeutic interventions [3,4]. The anatomically permissive environment of the ovary enables a distinctive characteristic of OC; tumor cells readily escape from the primary ovarian mass and disseminate throughout the peritoneal cavity at early stages of the disease [5]. OC cells exist within the peritoneal fluid as single cells or multicellular spheroids and develop anoikis resistance—a form of apoptosis evasion—that enables anchorage-independent survival [6]. Together with non-tumor cells in the peritoneal cavity, these malignant cells are propelled by peritoneal fluid flow and intestinal peristaltic movements, facilitating widespread dissemination to the pelvic organs, thereby enabling extra-pelvic and peritoneal colonization and metastases [6]. Thus, anoikis-directed therapeutic approaches are urgently needed to overcome the metastatic potential of ovarian cancer cells and to provide durable therapeutic responses in patients with advanced disease.

Ring-finger protein 126 (RNF126) is an E3 ubiquitin ligase composed of two zinc finger domains, namely, an N-terminal domain for substrate recognition and a C-terminal RING domain as the catalytic center, which coordinate the transfer of ubiquitin from E2 enzymes to target proteins [7,8]. RNF126 targets many proteins for ubiquitination and is involved in various biological processes, such as male fertility, quality control of misfolded proteins, energy metabolism, and cancer progression [9,10,11,12,13,14,15]. A previous study demonstrated that ERK-dependent upregulation of RNF126 in detached cells promotes ubiquitin-mediated degradation of pyruvate dehydrogenase kinases (PDKs), thereby liberating pyruvate dehydrogenase to enhance flux via the citric acid cycle and sustaining ATP production via oxidative phosphorylation. This metabolic reprogramming enables anchorage-independent cancer cell survival, whereas RNF126 depletion or PDK1 overexpression suppresses both anchorage-independent growth and tumorigenicity, highlighting the ERK-RNF126-PDK axis as a potential therapeutic target for disrupting metastatic dissemination in breast cancer [9]. In contrast, recent evidence has demonstrated that RNF126 is aberrantly upregulated in epithelial OC tissues and correlates with lymph node metastasis, pathological differentiation, and advanced FIGO stage, thus emerging as an independent biomarker for prognosis prediction [16]. Mechanistically, RNF126 promotes progression of OC by functioning as an E3 ubiquitin ligase that targets ArfGAP with coiled-coil, ankyrin repeat, and PH domains 2 (ACAP2) for proteasomal degradation, thereby reprogramming lipid metabolism and enhancing cancer cell proliferation, invasion, and migration both in vitro and in vivo [17]. However, whether and how RNF126 contributes to anoikis resistance and peritoneal colonization in OC remain unclear. To address this, we analyzed whether RNF126 depletion in OC cells affects anchorage-independent cell growth/anoikis resistance in vitro using ultra-low attachment culture plates as well as peritoneal colonization of OC cells in mouse models.

## 2. Results

### 2.1. RNF126 Depletion Promotes Anoikis in OC Cells

To determine the function of RNF126 in OC cells, we first established control and RNF126-depleted human serous ovarian cancer SKOV3 cells and mucinous ovarian cancer MCAS cells using the CRISPR/Cas9 system (Figure 1A,B). Although transient RNF126 knockdown by siRNA has been reported to attenuate proliferation of SKOV3 cells [16], stable RNF126 depletion did not affect the proliferation of SKOV3 or MCAS cells under normal culture conditions (i.e., adherent cells) (Figure 1C,D). Because RNF126 depletion did not affect the proliferation of OC cell lines under normal culture conditions for 72 h, we then examined whether RNF126 depletion affects anchorage-independent cell growth for 72 h. In contrast to normal culture conditions, RNF126 depletion decreased the cell number of SKOV3 and MCAS cells cultured under floating (anchorage-independent) conditions for 72 h (i.e., floating cells) by approximately 40–50% (Figure 1E,F). Thus, these results suggest that RNF126 depletion specifically attenuates anchorage-independent cell growth in both serous and mucinous OC cell lines.

Subsequently, we investigated whether RNF126 promotes anoikis in OC cells. OC cells were cultured under adherent conditions or floating conditions for 72 h, and dying and dead cells were stained with Ethidium homodimer III (EthD-III). We observed that only a small number of both control and RNF126-depleted cells under adherent conditions were EthD-III-positive; however, floating conditions induced a large number of SKOV3 and MCAS cells to become EthD-III-positive. Moreover, approximately 1.5 to 3 times more cells were positive for EthD-III in RNF126-depleted ovarian cancer cells compared to control cells (Figure 2A–D). These results indicate that RNF126 depletion promoted anoikis in OC cells.

### 2.2. RNF126 Depletion Suppresses Peritoneal Colonization of OC

Anoikis resistance is essential for metastatic dissemination and colonization in many types of cancers including OC [6,18,19]. Thus, we next explored whether RNF126 depletion affects the peritoneal colonization of ovarian cancer. SKOV3 and MCAS cells were injected intraperitoneally into immunodeficient mice, and peritoneal tumor nodules were collected 28 days after inoculation. RNF126 depletion significantly suppressed peritoneal colonization of both SKOV3 and MCAS cells (Figure 3A–D). Thus, these results imply that RNF126 is essential for peritoneal colonization of OC.

### 2.3. The Intact RING Domain Is Necessary for RNF126-Mediated Anoikis Resistance and Peritoneal Dissemination of Ovarian Cancer

RNF126 has one C3H2C3-type RING domain at the C-terminus, and the conserved C3H2C3 residues forms two Zn2+ ion-binding sites [7,20]. Thus, we investigated whether the intact C3H2C3 motif in the RING domain is necessary for RNF126-mediated anoikis resistance and peritoneal dissemination in SKOV3 cells. V5-tagged wild-type (WT) RNF126 or its mutant with an alanine substitution at the histidine 252 residue (H/A) was re-expressed in RNF126-depleted (sgRNF126#1) SKOV3 cells (Figure 4A). WT and H/A RNF126 re-expression did not affect cell proliferation under normal culture conditions (Figure 4B). However, WT but not H/A RNF126 restored anchorage-independent cell growth in RNF126-depleted SKOV3 cells (Figure 4C). EthD-III-positive dying/dead cells were also decreased in WT but not in H/A mutant RNF126 revertant cells under floating conditions (Figure 4D,E). Consistent with these in vitro results, WT RNF126 restored the peritoneal colonization ability of RNF126-depleted SKOV3 cells, but H/A RNF126 did not (Figure 4F,G). Taken together, these results suggest that the intact RING domain is essential for RNF126-mediated anoikis resistance and peritoneal colonization in OC.

### 2.4. RNF126 Activates NF-κB in Floating OC Cells

RNF126 is a ubiquitin E3 ligase, and its various substrates have been reported. Among these, p21 and PDK1 are related to cancer cell proliferation and colony formation ability in breast cancer and are regulated via proteasomal degradation by RNF126 [9,21]. In addition, Wang et al. have recently reported that ACAP2 degradation by RNF126 regulates proliferation in OC via alterations in lipid metabolism [17]. Therefore, we first analyzed protein levels of p21, PDK1, and ACAP2 in control and RNF126-depleted SKOV3 and MCAS cells under attached and floating conditions. Surprisingly, RNF126 depletion did not result in the accumulation of these proteins in SKOV3 and MCAS cells under either attached or floating conditions (Appendix A), implying that RNF126 promotes anchorage-independent growth of OC via unknown mechanism(s).

Next, we examined whether and how RNF126 depletion alters gene expression in SKOV3 cells under floating conditions. RNA-seq analysis revealed that 226 genes were upregulated and 929 genes were significantly down-regulated, with changes of 2-fold or more in RNF126-depleted cells (Appendix A). These differentially expressed genes (DEGs) were subjected to enrichment analyses using the Metascape algorithm. Gene ontology analyses showed that ontologies for drug metabolism and detoxification were enriched in the up-regulated DEGs, while those for development, differentiation, and inflammation were enriched in the DEGs down-regulated by RNF126 depletion (Appendix A). Employing TRRUST analyses, we found that genes regulated by TWIST1/2 were enriched in the upregulated DEGs, whereas those regulated by RELA and NFKB1 were enriched in the DEGs downregulated by RNF126 depletion (Appendix A). The NF-κB signaling pathway is a well characterized pathway in anoikis resistance of cancer [22,23,24,25,26]. Thus, we further focused on the relationship between RNF126 and NF-κB signaling.

To monitor NF-κB activity in OC cells, NF-κB reporter plasmids were transfected into control and RNF126-depleted SKOV3 and MCAS cells. RNF126 depletion did not affect NF-κB activity in OC cells under attached conditions (Figure 5A,B; attached). However, when cells were cultured under floating conditions, the NF-κB activity was increased by approximately 1.5 to 3 times compared with that under attached conditions in control cells (Figure 5A,B; floating). In addition, this increase was completely abolished by RNF126 depletion (Figure 5A,B). Thus, these results indicate that RNF126 activates NF-κB activity in floating OC cells. It has been reported that in the NF-κB canonical pathway, IκB degradation liberates cytosolic NF-κB p65/p50, and nuclear translocated p65/p50 functions as a transcriptional factor [27,28]. Because RNF126 activates NF-κB activity in floating OC cells, we examined whether RNF126 affects the nuclear accumulation of p65. Interestingly, in our study, we observed that nuclear p65 accumulation was abolished by RNF126 depletion under floating conditions without affecting IκBα protein levels in whole cell lysates of SKOV3 and MCAS cells (Figure 5C–F), implying that RNF126 activates NF-κB independently of IκB degradation in floating OC cells. This RNF126-mediated NF-κB activation in OC under floating conditions also needed the presence of the intact RING domain (Figure 5G–I), similar to RNF126-mediated anoikis resistance and peritoneal colonization.

### 2.5. NF-κB Activity Is Essential for RNF126-Mediated Anchorage-Independent Growth and Peritoneal Colonization in Ovarian Cancer

Finally, we checked whether NF-κB activity is indeed necessary for RNF126-mediated anchorage-independent growth in OC. RNF126-mediated anchorage-independent growth was abolished in SKOV3 and MCAS cells when the cells were treated with the NF-κB inhibitor, QNZ, at concentrations that did not show severe toxicity in OC cells under attached conditions (Figure 6A–D). In addition, administration of only QNZ for the first 3 days after peritoneal dissemination significantly decreased peritoneal SKOV3 tumor nodules by approximately 50% (Figure 6E,F). Thus, NF-κB activity is essential for RNF126-mediated anchorage-independent growth and peritoneal colonization of cells in OC.

## 3. Discussion

Peritoneal dissemination and colonization greatly affect the prognosis of OC, and therapy for peritoneally disseminated and colonized OC remains limited. Here, we showed that RNF126 contributes to anchorage-independent growth and peritoneal dissemination of cells in OC. RNF126 activates NF-κB in floating OC cells, thereby promoting anchorage-independent growth depending on the intact RING domain of RNF126. The intact RING domain of RNF126 is also necessary for RNF126-mediated anoikis resistance in OC. In parallel with these in vitro observations, RNF126 depletion and NF-κB inhibitor administration significantly suppressed the peritoneal colonization of OC in mouse models.

A previous study has shown that transient knockdown of RNF126 by siRNA attenuates proliferation of SKOV3 cells [16]. However, our experiments showed that stable knockout of RNF126 using the CRISPR/Cas9 system did not affect the proliferation of SKOV3 and MCAS cells under normal culture conditions. A similar discrepancy between transient and stable knockdown/knockout of RNF126 on cell proliferation has been reported in breast cancer [9,21]. This may be explained by the possibility that other molecules compensate for the cell proliferation defects caused by long-term RNF126 depletion. In contrast to cell proliferation, the abilities of anchorage-independent cell growth/anoikis resistance and peritoneal colonization of cells remained suppressed by long-term RNF126 depletion in OC. Thus, RNF126 inhibitors may suppress peritoneal colonization for a long period without activating pathways that bypass RNF126 in OC. Recently, Wang et al. have reported that stable knockdown of RNF126 by shRNA attenuated cell proliferation due to the accumulation of ACAP2 protein, resulting in downregulation of lipid metabolism-related genes in OC A2780 cells [17]. However, we were unable to detect accumulation of the ACAP2 protein in RNF126-depleted SKOV3 and MCAS cells under either attached or floating conditions. In addition, RNA-seq analyses did not show significant differences in the expression levels of lipid metabolism-related genes such as FASN, ACC, and SCD, between the floating control and RNF126-depleted SKOV3 cells. Whether this discrepancy is attributable to differences in cell lines, method of gene suppression, or other reasons remains unclear.

During peritoneal dissemination and colonization, cancer cells seeded from the primary site should be able to survive in the peritoneal cavity; attach to the mesothelium; and invade into, colonize, and grow in the peritoneum [6,29,30]. NF-κB can contribute to multiple steps of peritoneal dissemination. For example, RelA/p65 enhances ovarian tumorigenesis and metastasis by supporting proliferation [31]. NF-κB promotes invasion via epithelial–mesenchymal transition in OC [32]. Although RNF126-mediated NF-κB activation is observed under floating conditions in OC cells, we cannot exclude the possibility that RNF126-mediated NF-κB activation promotes not only anchorage-independent growth/anoikis resistance in the peritoneal cavity but also other steps of peritoneal colonization in OC.

A limitation of this study is that detailed mechanism(s) underlying RNF126-mediated NF-κB activation remain unclear. The RNF126 revertant experiments indicate that the intact RING domain is necessary for NF-κB activation by RNF126. The RING domain is essential for the ubiquitination property of RNF126 and other RING-type ubiquitin ligases [20,33]. Previous reports have shown that RNF126 can ubiquitinate various proteins via K48- and K63-linked chains [8,9,10,21,34,35]. Interestingly, IκBα protein levels were not affected in RNF126-depleted OC cells during the floating conditions. Thus, these results reveal that RNF126 targets IκBα degradation-independent mechanisms of NF-κB activation in OC. Various IκBα degradation-independent NF-κB activation mechanisms have been reported: Ser180 phosphorylation of p65 by CLK2 promotes nuclear export of p65 [36], p53 activates NF-κB via p65 phosphorylation by RSK1 [37], tyrosine phosphorylation of IκBα releases IκBα from NF-κB without IκBα degradation [38]. We have not addressed the involvement of the above-mentioned mechanisms in RNF126-mediated NF-κB activation in OC; however, the p53-mediated pathway is unlikely to be involved, because SKOV3 cells do not express endogenous p53 [39]. In addition to NF-κB-related proteins, sensor molecules for cell adhesion, such as integrins, cytoskeletons, and their regulators, might be targets of RNF126 because RNF126-mediated NF-κB activation occurs only under floating conditions of cells in OC. This study did not identify the direct substrate of RNF126 that regulates NF-κB, but combined with the E3 ubiquitin ligase function of RNF126 and existing NF-κB activation pathways, it is speculated that RNF126 may target the ubiquitination modification of p65 or upstream signaling molecules. Future studies employing proteome analyses may reveal RNF126-mediated ubiquitinated/degraded proteins and detailed mechanisms by which RNF126 activates NF-κB during the floating conditions of cells in OC.

In summary, RNF126 depletion suppresses NF-κB–dependent anchorage-independent growth and peritoneal colonization in OC. Our findings indicate that RNF126 is a possible target for the prevention and/or therapy of peritoneally disseminated and colonized OC.

## 4. Materials and Methods

### 4.1. Cell Culture

SKOV3 cells were obtained from the American Type Culture Collection (Manassas, VA, USA). MCAS cells were obtained from the Japanese Collection of Research Bioresources Cell Bank (Osaka, Japan). The cells were cultured in RPMI 1640 medium (FUJIFILM Wako, Osaka, Japan) containing 10% fetal bovine serum (FBS), 100 units/mL penicillin, and 100 µg/mL streptomycin at 37 °C in a humidified incubator with 5% CO_2_. All the cell lines were routinely tested to exclude mycoplasma contamination.

### 4.2. Vector Construction

The lentivirus construct, LentiCRISPRv2 [40,41], was obtained from Addgene (Cambridge, MA, USA). The targeted sequences for single guide RNA (sgRNA) were as follows: control sgRNA (sgCTR), 5′-GCGAGGTATTCGGCTCCGCG-3′; sgRNF126#1, 5′- CAGTGGCAGAAGTACCGTCC-3′; sgRNF126#2, 5′- ACGACGGCTGCATCGTGCCC-3′. Oligo DNA of the targeted sequences was subcloned into the LentiCRISPRv2 plasmid according to the manufacturer’s instructions as described previously [42]. Human RNF126 cDNA was amplified from SKOV3 cells using RT-PCR. Constructs expressing the wild-type and H252A mutant RNF126 with silent mutations at the sgRNF126#1 target site were prepared using a PCR-based method. These cDNAs were subcloned into the pENTR/D-TOPO vector (Thermo Fisher Scientific, Waltham, MA, USA) before being incorporated into the lentivirus vector pLenti6/V5 DEST (Thermo Fisher Scientific, Waltham, MA, USA), as described previously [43,44]. Lentiviral particles were produced by transient co-transfection of 293FT cells with LentiCRISPRv2 or pLenti6 plasmids and ViraPower Lentiviral Packaging Mix (Thermo Fisher Scientific) using the Lipofectamine 3000 reagent (Thermo Fisher Scientific), according to the manufacturer’s instructions.

### 4.3. Cell Growth Assay and Ethidium Homodimer III (EthD-III) Staining

Cells (1 × 10^4^ for attached conditions and 5 × 10^4^ for floating conditions) were seeded in normal plastic tissue culture plates for attached conditions and in ultra-low attachment plates for floating conditions (Corning Inc., Corning, NY, USA), and cultured at 37 °C in a humidified CO2 incubator for the indicated days with or without QNZ (100 nM; Tokyo Chemical Industry, Tokyo, Japan). The cells were counted periodically using a hemocytometer. EthD-III staining was performed as described previously [45]. Briefly, to detect cell death, cells cultured for 72 h under the indicated conditions were stained with Hoechst33342 (1 μg/mL; Merck Millipore, Burlington, MA, USA) for all cells and EthD-III (2 μM; PromoCell, Heidelberg, Germany) for dying or dead cells 15 min before observation under a fluorescent microscope (Keyence, Osaka, Japan). EthD-III fluorescence intensity was quantified using the ImageJ software (ver. 1.54p; National Institutes of Health, Bethesda, MD, USA) and normalized to Hoechst33342 fluorescence intensity.

### 4.4. Western Blotting Analysis

For total cell lysates, cells were lysed with lysis buffer (1% Nonidet P-40, 50 mM Tris pH 8.0, and 150 mM NaCl) and centrifuged at 20,000× *g* for 15 min at 4 °C. Next, the supernatants were collected. Nuclear lysates were prepared using the Nuclear Extract Kit (Active Motif, Carlsbad, CA, USA) according to the manufacturer’s instructions. Total protein content in the lysates was measured using the Bradford assay (Bio-Rad, Hercules, CA, USA). The lysates were separated using SDS-PAGE, transferred to polyvinylidene fluoride membrane filters, and analyzed by Western blotting using mouse anti-β-actin antibody (MAB1501; Merck Millipore, Burlington, MA, USA), mouse anti-RNF126 antibody (66647-1-Ig; Proteintech, Rosemont, IL, USA), mouse anti-IκBα antibody (4814; Cell Signaling Technology, Danvers, MA, USA), mouse anti-Lamin A/C (SC-7292; Santa Cruz Biotechnology, Dallas, TX, USA), mouse anti-p65 antibody (6956; Cell Signaling Technology, Danvers, MA, USA), rabbit anti-PDK1 antibody (LS-C26009; LifeSpan BioSciences, Newark, CA, USA), rabbit anti p21 antibody (2947; Cell Signaling Technology), rabbit anti ACAP2 antibody (14029-1-AP; Proteintech, Rosemont, IL, USA), horseradish peroxidase (HRP)-linked anti-rabbit IgG antibody (7074; Cell Signaling Technology), and HRP-linked anti mouse IgG antibody (7076; Cell Signaling Technology). Chemiluminescence was detected using the ImageQuant LAS400 mini imager (Cytiva, Marlborough, MA, USA) and the ECL Select/Prime reagent (Cytiva, Marlborough, MA, USA).

### 4.5. Peritoneal Dissemination of Ovarian Cancer in Mice

Mice were maintained under specific pathogen-free conditions. Experimental protocols were approved by the Animal Care and Use Committee of Kansai Medical University (No. 24-51 and 25-025), and the experiments were conducted according to the ARRIVE guidelines 2.0, the institutional ethical guidelines for animal experiments, and safety guidelines for gene manipulation experiments. The sample size was based on statistical analysis of variance and exploratory experiments (10 mice/group for SKOV3 cells and 7 mice/group for MCAS cells). BALB/c nude mice were purchased from Clea Japan (Tokyo, Japan), and mice were randomly assigned to each experimental group. Peritoneal dissemination of the cells was examined using 8-week-old female BALB/c nude mice. Cells cultured on normal dishes were collected for peritoneal injection. Cells (1 × 10^6^) in 0.2 mL phosphate-buffered saline were injected intraperitoneally into mice. For QNZ administration experiments, vehicle (phosphate-buffered saline) or QNZ (200 µg/kg body weight) was administered daily for the first three days after tumor inoculation. At the endpoint (Day 28), peritoneal tumor nodules were collected, weighed, and photographed. All mice were included in the analysis. Investigators were not blinded during data acquisition and analysis.

### 4.6. RNA-Seq and Informatics Analysis

Control (sgCTR) and RNF126-depleted (sgRNF126#1) SKOV3 cells were cultured under floating conditions for 24 h, and total RNA was isolated from the cells (three biological replicates per group) using the RNeasy Mini Plus kit (Qiagen, Hilden, Germany). Library preparation (NEBNext Ultra II Directional RNA Library Prep Kit; New England Biolabs, Ipswich, MA, USA) and sequencing (DNBSEQ-T7 platform, paired-end 150 bp; MGI Tech, Shenzhen, China) were performed by Novogene (Beijing, China). Transcript-level quantification was performed using Kallisto [46]. The estimated transcript abundance was imported into the R software (ver. 4.2.1), using the tximport package (ver. 1.26.1) and summarized as gene-level expression values. For genes with multiple transcript isoforms, the transcript with the highest overall expression was selected as a representative transcript. Normalized gene expression values (scaledTPM) were used for subsequent analyses. Specifically, gene expression between control and RNF126-depleted SKOV3 cells was analyzed using the DESeq2 R package (1.38.3), and genes with a Benjamini–Hochberg adjusted *p*-value < 0.05 and ≥2-fold difference were selected. Among these, the upregulated or downregulated genes were subjected to gene ontology and TRRUST analyses using the Metascape algorithm [47]. RNA-seq data were deposited in the DDBJ databases (https://www.ddbj.nig.ac.jp/index-e.html, accessed on 18 December 2025) (Accession number: PRJDB39887).

### 4.7. Reporter Assay

NF-κB activity was measured using pGL4.32 vectors that express firefly luciferase under the control of five copies of an NF-κB response element (Promega, Madison, WI, USA). A pRL vector expressing Renilla luciferase (Promega) served as an internal control. Cells (1 × 10^6^) were seeded in 10 cm dishes and co-transfected with a reporter plasmid (5 µg) and internal control pRL vector (500 ng) using Lipofectamine 3000 (Thermo Fisher Scientific). Next day, the cells were re-seeded in normal and ultra-low attachment plates at 1 × 10^5^ cells/mL, and 24 h after re-seeding, luciferase activity of lysates of the transfected cells was measured using the Dual Luciferase Reporter Assay System (Promega), according to the manufacturer’s instructions. Luminescence was measured using the GloMax 20/20 Luminometer (Promega).

### 4.8. Statistics

The Brown-Forsythe and Welch ANOVA tests and unpaired *t*-test with Welch’s correction were used for statistical evaluations using GraphPad Prism software (ver.10.4.1; GraphPad Software, Inc., La Jolla, CA, USA). Data are presented as the mean ± SEM or ± SD; The mean is labeled on the graph; *p*-values < 0.05 were considered significant. Sample size is based on statistical analysis of variance and on exploratory experiments. Each in vitro experiment was replicated at least three times successfully. Animal experiments were replicated at least twice successfully.

## Figures and Tables

**Figure 1 ijms-26-12183-f001:**
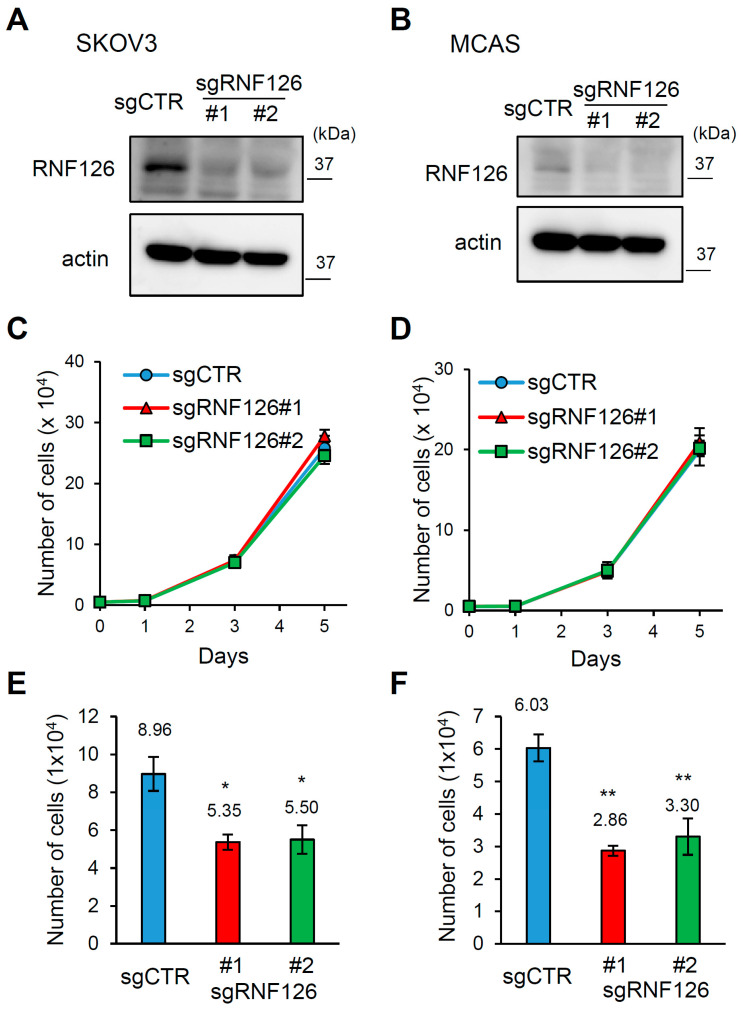
RNF126 depletion attenuates anchorage-independent cell growth in ovarian cancer cells. (**A**,**B**) Immunoblotting of RNF126 in control (sgCTR) and RNF126-depleted (sgRNF126#1, #2) SKOV3 (**A**) and MCAS cells (**B**). (**C**,**D**) Growth of control and RNF126-depleted SKOV3 (**C**) and MCAS (**D**) cells cultured under normal conditions (adherent cells). The initial number of cells seeded is 1 × 10^4^. (**E**,**F**) Cell number of control and RNF126-depleted SKOV3 (**E**) and MCAS (**F**) cells cultured for 72 h under floating conditions. The initial number of cells seeded is 5 × 10^4^. In (**C**–**F**), data represent mean ± SD (n = 3). * *p* < 0.05, ** *p* < 0.01 using the Brown-Forsythe and Welch ANOVA tests.

**Figure 2 ijms-26-12183-f002:**
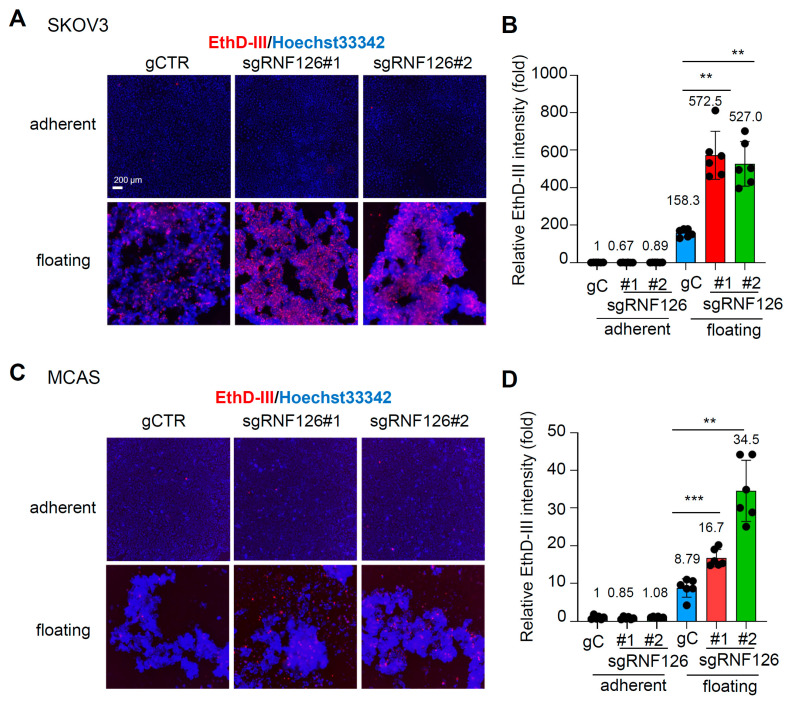
RNF126 depletion promotes anoikis in ovarian cancer cells. (**A**–**D**) Dying or dead cells were stained with Ethidium homodimer III (EthD-III) dye (red) and nuclei were stained with Hoechst33342 dye (blue). (**A**,**C**) Representative photos of EthD-III- and Hoechst33342-stained SKOV3 (**A**) and MCAS (**C**) cells cultured under the indicated conditions. Scale bar: 200 μm. (**B**,**D**) EthD-III fluorescence intensity normalized to Hoechst33342 intensity was measured. Bars in blue, red, and green represent gCTR, sgRNF126 #1, and sgRNF126 #2 cells. Data represent mean ± SD (n = 6). ** *p* < 0.01, *** *p* < 0.001 using the Brown-Forsythe and Welch ANOVA tests.

**Figure 3 ijms-26-12183-f003:**
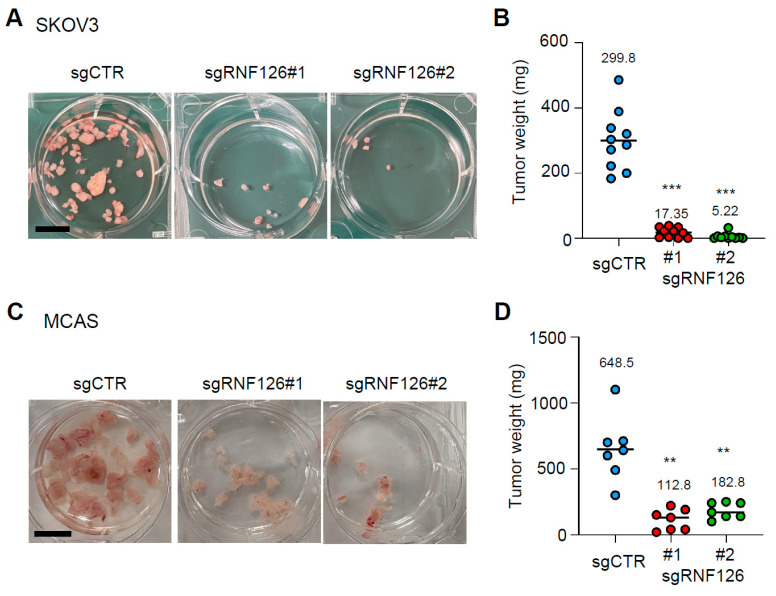
RNF126 depletion suppresses peritoneal colonization of ovarian cancer cells. (**A**,**C**) Representative photos of resected peritoneal tumor nodules of SKOV3 (**A**) and MCAS (**C**) cells. Scale bar: 1 cm. (**B**,**D**) Weights of resected peritoneal tumor nodules of SKOV3 ((**B**), n = 10 from two independent experiments) and MCAS ((**D**), n = 7 from two independent experiments) cells were measured. ** *p* < 0.01 and *** *p* < 0.001 using the Brown-Forsythe and Welch ANOVA tests.

**Figure 4 ijms-26-12183-f004:**
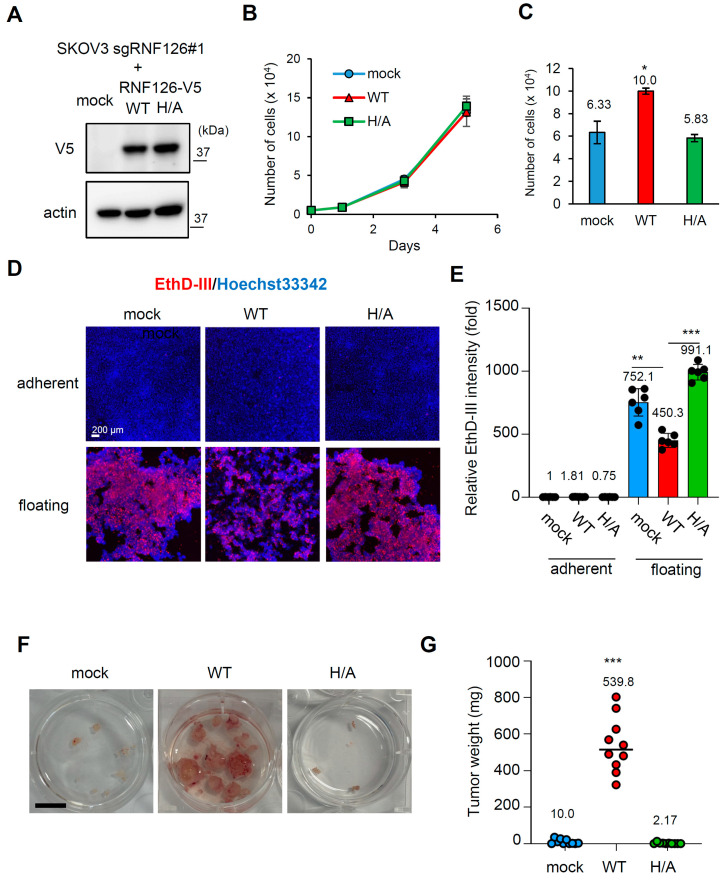
Intact RING domain is necessary for RNF126-mediated anoikis resistance and peritoneal colonization in SKOV3 cells. (**A**) Immunoblotting of mock, V5-tagged wild-type (WT), and H252A mutant (H/A) RNF126 in RNF126-depleted (sgRNF126#1) SKOV3 cells. (**B**) Growth of RNF126 revertant SKOV3 cells in normal culture conditions. Data represent mean ± SD (n = 3). The initial number of cells seeded is 1 × 10^4^. (**C**) Cell number of RNF126 revertant SKOV3 cells cultured for 72 h under floating conditions. The initial number of cells seeded is 5 × 10^4^. Data represent mean ± SD (n = 3). (**D**) Representative photos of EthD-III- and Hoechst33342-stained SKOV3 cells cultured under the indicated conditions. Scale bar: 200 μm. (**E**) EthD-III fluorescence intensity normalized to Hoechst33342 intensity was measured. n = 6. (**F**) Representative photos of resected peritoneal tumor nodules of RNF126 revertant SKOV3 cells. Scale bar: 1 cm. (**G**) Weights of the resected peritoneal tumor nodules of SKOV3 cells were measured. n = 10 from two independent experiments. In (**C**,**E**,**G**), * *p* < 0.05, ** *p* < 0.01, *** *p* < 0.001 using the Brown-Forsythe and Welch ANOVA tests.

**Figure 5 ijms-26-12183-f005:**
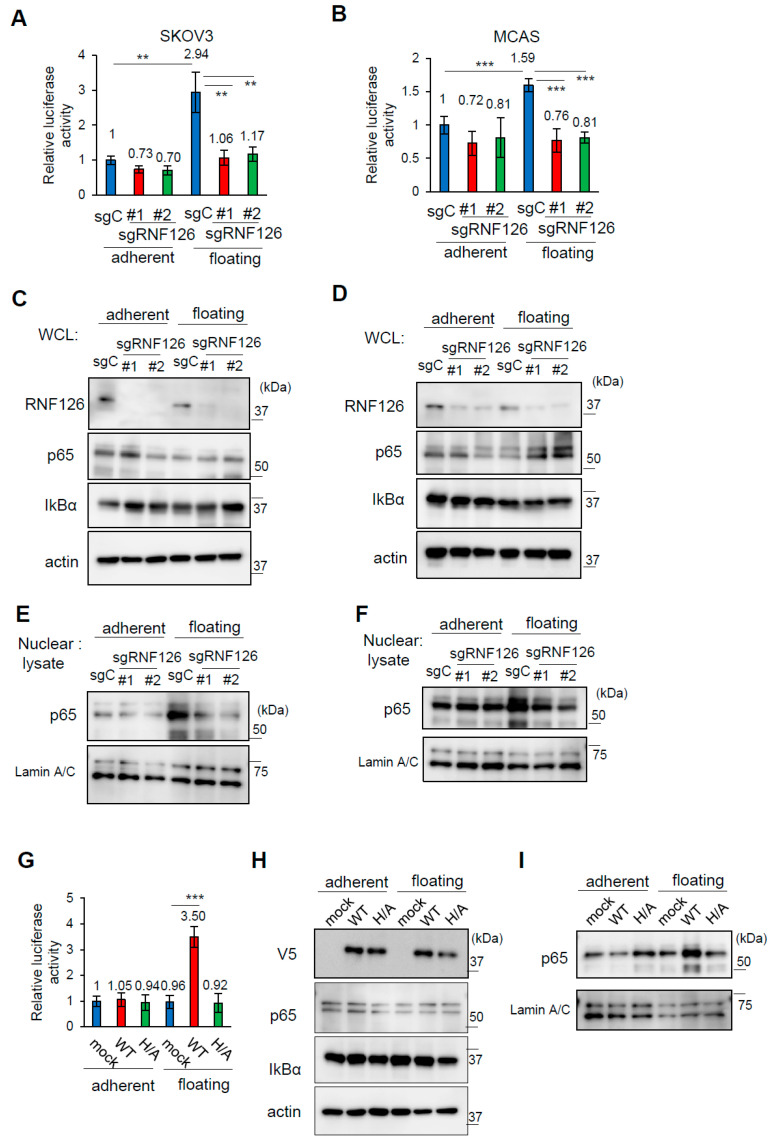
RNF126 depletion attenuates NF-κB activity in floating ovarian cancer cells. (**A**,**B**) Luciferase activity of NF-κB reporters in control (gC) and RNF126-depleted (sgRNF126#1, #2) SKOV3 (**A**) and MCAS (**B**) cells cultured under the indicated conditions. n = 5. (**C**,**D**) Immunoblotting of RNF126, NF-κB p65, and IκB in whole cell lysates (WCL) of control and RNF126-depleted SKOV3 (**C**) and MCAS (**D**) cells. (**E**,**F**) Immunoblotting of NF-κB p65 in nuclear lysates of control and RNF126-depleted SKOV3 (**E**) and MCAS (**F**) cells. Lamin A/C is a marker of nuclear protein. (**G**) Luciferase activity of NF-κB reporters in RNF126 revertant SKOV3 cells cultured under the indicated conditions. (**H**,**I**) Immunoblotting of RNF126, NF-κB p65, and IκBα in WCL (**H**) and nuclear lysates (**I**) of RNF126 revertant SKOV3 cells. In (**A,B,G**), data represent mean ± SD (n = 5). ** *p* < 0.01, *** *p* < 0.001 using the Brown-Forsythe and Welch ANOVA tests.

**Figure 6 ijms-26-12183-f006:**
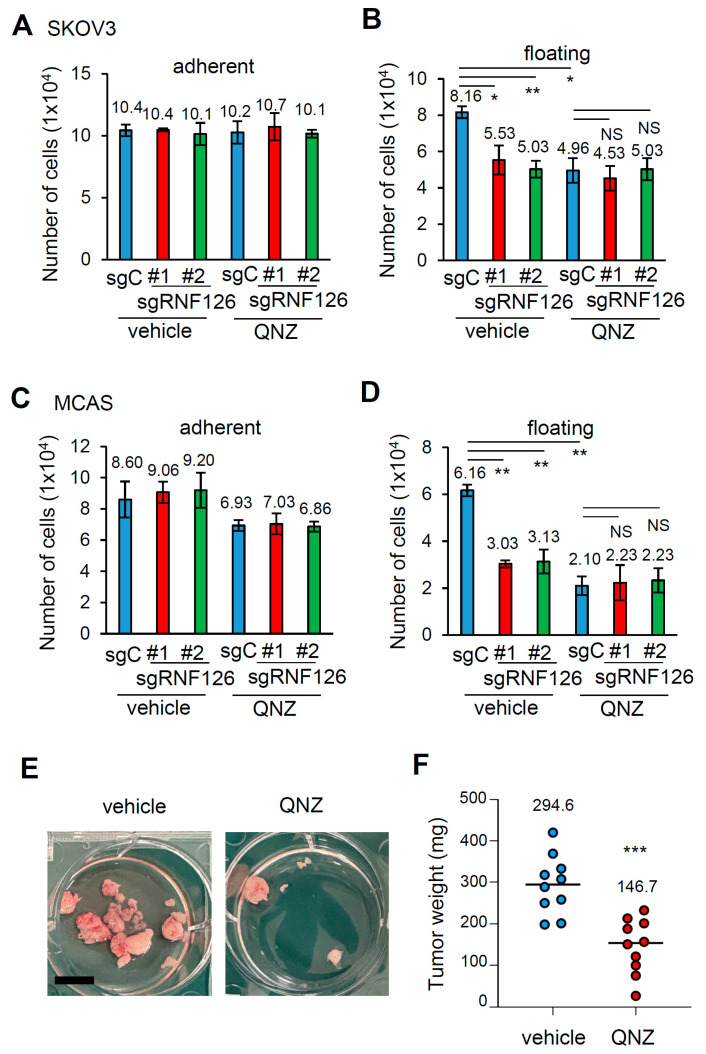
NF-κB activity is necessary for RNF126-mediated anchorage-independent growth and peritoneal colonization in SKOV3 cells. (**A**–**D**) Cell number of control (sgC) and RNF126-depleted (sgRNF126#1, #2) SKOV3 (**A**,**B**) and MCAS (**C**,**D**) cells cultured for 72 h under attached (**A**,**C**) and floating (**B**,**D**) conditions with or without NF-κB inhibitor QNZ (100 nM). Data represent mean ± SD (n = 3). * *p* < 0.05, ** *p* < 0.01, NS; not significant using the Brown-Forsythe and Welch ANOVA tests. (**E**,**F**) QNZ (200 ng/kg body weight) was administered daily for the first three days following SKOV3 inoculation. Representative photos of resected peritoneal SKOV3 tumor nodules (**E**). Weights of the resected peritoneal SKOV3 tumor nodules (**F**). n = 10 from two independent experiments. *** *p* < 0.001 using unpaired *t*-test with Welch’s correction.

## Data Availability

The original contributions presented in this study are included in the article/Appendix A. Further inquiries can be directed to the corresponding authors.

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
