# Peer review of "Ring-Finger Protein 126 (RNF126) Promotes Anoikis Resistance and Peritoneal Colonization in Ovarian Cancer"

_ijms, 2025, doi:10.3390/ijms262412183_

Round 1
Reviewer 1 Report
Comments and Suggestions for Authors
This manuscript focuses on the role of RNF126 in anoikis resistance and peritoneal metastasis of ovarian cancer, and identifies the "RNF126-RING domain-NF-κB" regulatory axis. The research direction holds significant clinical translational value. The experimental design is reasonable, the data support is reliable, and no substantive flaws in the core conclusions have been found. However, minor revisions are required regarding expression standardization and logical rigor, and the manuscript can be directly accepted after revision.
Minor Revision Suggestions:
1.Visualization of Graphical Data Presentation
Suggestion: Supplement a volcano plot of RNA-seq differentially expressed genes (DEGs) (which can be generated based on existing Table S1 data) to intuitively show the distribution characteristics of upregulated/downregulated genes and enhance data visualization.
2. Enhancing the Rigor of Mechanistic Descriptions
The description of "the specific way RNF126 activates NF-κB" in the discussion is not sufficiently rigorous. Only some known substrates have been excluded, but no reasonable speculation has been made in combination with existing studies, leading to ambiguity in the mechanistic part.
Suggestions:
- Supplement reasonable speculation on the IκBα-independent NF-κB activation mechanism in the discussion. Incorporate existing literature (such as p65 phosphorylation and integrin signaling mentioned in the manuscript) to illustrate that RNF126 may function by regulating the ubiquitination modification of NF-κB pathway-related molecules (e.g., activation via K63-linked ubiquitination). No additional experiments are needed; only logical connections need to be strengthened.
- Clearly state that "this study did not identify the direct substrate of RNF126 that regulates NF-κB, but combined with the E3 ubiquitin ligase function of RNF126 and existing NF-κB activation pathways, it is speculated that RNF126 may target the ubiquitination modification of p65 or upstream signaling molecules" to avoid absolute expressions.
3. Format Standardization
(1) Inconsistent use of abbreviations: The full names of abbreviations such as "FIGO", "DEG", and "EthD-III" were not indicated upon their first appearance in the main text.
Suggestion: Uniformly standardize the indication of full names for all abbreviations when they first appear.
(2) Figure 5 is excessively large, affecting the aesthetics of the manuscript layout.
Suggestion: Present Figure 5 as a supplementary figure.
4. Completeness of Supplementary Data Annotations
(1) The annotations for some experimental data are insufficiently detailed. For the "number of cells after 72 hours of suspension culture" in Figure 1E and F, the cell seeding density (5×10⁴ cells/well as mentioned in the Materials and Methods section) was not specified in the figure legends, making it difficult for readers to judge the baseline consistency of growth differences.
Suggestion: Supplement key experimental parameters in the legends of relevant figures and tables, such as cell seeding density and drug treatment concentration (e.g., 100 nM for QNZ), to ensure experimental reproducibility.
(2) Upload raw sequencing data to public databases
Suggestion: Upload the raw RNA-seq sequencing reads (in fastq format) to internationally recognized public databases, such as NCBI SRA, ENA, or GEO, to obtain an Accession Number.
Reviewer 2 Report
Comments and Suggestions for Authors
The manuscript ‘Ring-finger protein 126 (RNF126) promotes anoikis resistance and peritoneal dissemination in ovarian cancer’ presents an interesting set of in vitro to in vivo experiments on a protein that may be a potent target in ovarian cancer. Unfortunatey it needs major rewriting of the methods sections 8major details are missing) and clarifications in of the results presentation.
- In general it would be helpful, if the results would be supported by numbers in the text.
- Despite impressive effects in a valid mouse model, it is not a dissemination model, but a colonization model. As the authors describe dissemination as tumor cells being released from the ovary to the peritoneal cave. Here cell line cells are injected in ‘large’ numbers , omitting the first crucial steps in dissemination. Which cells have been used (attached or floating?) Tumor weight per mouse is given in point graphs, but numbers are missing. In addition the authors state, that the mouse experiments have been done at least twice. Data should be shown, since n=5 is a very low group size (even though might be justified by the large effect). Numbers of nodules/per mouse are missing, as is histology.
- Regarding the Western blots, experimental details e. d. secondary antibodies and detection methods, stripping are missing. Since anoikis/detachment results in cytoskeleton changes, it should be explained, why actin has been used as a housekeeper. Bar graphs with quantification would be helpful. Phosphorylation of Ser180 is described as an important activation (and translocation) mechanism for p65, yet it has not been analysed. Please comment.
- In Figure 1 a time course with 3 timepoints is shown for adhesive growth, yet for the floating cells and all follow up cell culture experiments, only 72h are shown. The authors should give an explanation for this and the choice of timing.
- Unfortunatey the images in Figure 2 are only blue blobs in the manuscript version provided to this reviewer (too small with too low resolution)
Round 2
Reviewer 2 Report
Comments and Suggestions for Authors
The revised manuscript ‘Ring-finger protein 126 (RNF126) promotes anoikis resistance and peritoneal dissemination in ovarian cancer’ has improved by giving more details on the methods, improved fluorescence images and reanalysis of the animal experiment(s) giving data on all animal tested.
Major concerns remain:
- The authors have included some numbers in the text for the effects shown in the Figures.
Yet e. g. for Figure 1: ‘approximately 40-50%’ is not a statement for the scope of this manuscript (to be expected: mean/SD as given in the graph/figure legends).
The methods description has improved. It would be helpful, if the authors would include the description of the nodules (clumps of nodules, lack of material in RNF126 ko, because of ‘treatment effect’). This reviewer acknowledges difficulties is assessing number and sizes of multiple nodules, and total tumor load as a good first parameter, but valuable information is lost.
- Here seems to be a misunderstanding for the peritoneal tumor model. This reviewer did not question the model, cell injection (including cell number), but wanted to point out, that this is a (valid) colonization/homing model for peritoneal carcinosis, but not a dissemination model, as follows by the authors description of ‘feeing cells from the ovary’ beeing the cruzial first step. A similar discussion occurs regularly when discussing metastasis models.
3.Quantification of WB is a wide field, constantly improving.
Protein concentration by BCA is – as the authors state – not an appropriate measure for protein content on the WB membrane. It is the reason for loading control by measuring a housekeeper (as actin) on the same membrane as the analyte of interest or use a (commercially available) ‘whole protein stain for membranes’.
Since the authors state in their comments, that ‘membranes were not stripped horizontally’ and the ‘original WB’ show different smilies, the WB are not state of the art and lack comparability. This issue is critical and needs to be resolved.
This reviewer apologizes for not realizing, that not p65-pSer180 is available and acknowledges the authors to include a statement on ‘speculating’ on p65-P180 as a mechanism. In this context, some more explanation for using nuclear extracts (and Lamin A/C) would be helpful.
- In Figure 1 a time course with 3 timepoints is shown for adhesive growth, yet for the floating cells and all follow up cell culture experiments, only 72h are shown. The explain the choice of timing. Yet the results, withno difference in normal condition, but significant changes under floating conditions, make a time course for ‘floating cells’ the more interesting.
- Thank you for improving the images in Figure 2. Now a new question comes up:
Are there morphological differences between the cells under floating conditions? In Figure 2A gCTR looks more like a layer, the other two more 3D. Same impression is in Figure 4D with WT being the ‘flat looking’ one.
Round 3
Reviewer 2 Report
Comments and Suggestions for Authors
I accept that the authors did their best to answer my comments - and
came in some parts to their limits.
With the explanations/changes and the scope of this journal I would
accept the manuscript - execpt for my concerns on the Western Blots,
that cannot be solved.
Here I have to leave it to the editors, if using different membranes
(with inherent loading errors) for analyte and housekeeper is acceptable
or not.